# Retrieving characteristics of Inertia Gravity Wave parameters with least uncertainties using hodograph method

Gopa Dutta[1], P. Vinay Kumar[1] and Salauddin Mohammad[1]

[1]Vignana Bharathi Institute of Technology, Hyderabad 501301, India.

*Correspondence to*: Gopa Dutta (gopadutta@yahoo.com)

**Abstract.** We have analyzed wind velocities measured with high resolution Global Positioning System (GPS) radiosondes which have been flown continuously for 120 h with an interval of 6 h from Hyderabad. Hodograph method has been used to retrieve the Inertia Gravity Waves (IGW) parameters. Background winds are removed from the time series by detrending whereas polynomials of different orders are removed to get the fluctuations from individual profiles. Butterworth filter is

used to extract monochromatic IGW component. Another filter Finite Impulse Response (FIR1) is tried in a similar manner to test the effects of filters in estimating IGW characteristics. Results reveal that the fluctuation profiles differ with the change of polynomial orders, but the IGW parameters remain same when Butterworth filter is chosen to extract the monochromatic wave component. The FIR1 filter produces results with a broader range. The direction of wave propagation can be confirmed with additional temperature information.

**1 Introduction**

It is well documented that gravity waves of different scales play an important role in maintaining the large-scale circulation of the middle atmosphere. A number of studies have been carried out to characterize these waves by using different techniques. A very common, established and standard procedure of characterizing Inertia Gravity Waves (IGW) with frequencies close to Coriolis frequency is by hodograph method (Guest et al., 2000; Ogino et al., 2006; Niranjan Kumar et

al., 2011). Radiosonde data of horizontal winds and temperature have been extensively used to study these waves (Tsuda et al., 2004; Vincent and Alexander, 2000; Gong et al., 2008; Chane-Ming et al., 2010, 2014; Murphy et al., 2014; Kramer et al., 2015). Nastrom and VanZandt (1982) reported good accuracy in gravity wave parameters derived using balloon measurements since balloons have good aerodynamic responses. In a simulation study Wei and Zhang (2014) have demonstrated that gravity waves with different frequencies and generated by different sources like jet-imbalance and

convection can coexist together. The popular hodograph method demands the presence of a single coherent wave in the fluctuation profiles and does not yield good result when a mixture of various frequencies are present. The gravity wave parameters extracted by hodograph method might also be inaccurate when multiple waves are present in the data (Eckermann and Hocking, 1989).

Hodograph method is based on linear theory of gravity waves whereas the dynamics of the flow is more complex and non-

linear which introduces some uncertainties in their interpretations. There are several sources of errors in this method which have been described in Zhang et al., (2004). These authors compared the gravity wave characteristics obtained using hodograph method with the values derived from 4D output of their simulation study. A narrow bandwidth filter used by them to extract the fluctuations of a near-monochromatic wave resulted in large uncertainties in the horizontal wavelength which got reduced for waves with shorter vertical wavelengths. Even the spatial variations of the wave characteristics were found

to be large. Moreover, since the hodographs are quite variable, a large number of hodographs (profiles) are required to get

accurate results of gravity wave parameters with some statistical significance (Hall et al., 1995). This limits the very advantage of hodograph method which is used to retrieve GW parameters from a single set of vertical profiles of zonal and meridional winds.

In this study, we have attempted to reduce uncertainties associated with hodograph method in delineating the characteristics of IGW from wind velocities obtained with radiosonde measurements.

## 2 Experiment and Data

An intensive campaign with high resolution (i-Met, USA) GPS-radiosonde flights was carried out from the campus of India Meteorological Department (IMD), Hyderabad (17.4 °N, 78.5 °E) with four flights a day at an interval of 6 h for 5 consecutive days (20 flights) between 30 April and 4 May, 2012 to study the characteristics of IGW. The timings of the flights were 05:30, 11:30, 17:30 and 23:30 LT. The accuracy of wind and temperature measurements is ±1 ms$^{-1}$ and ± 0.2 K respectively (Vinay Kumar et al., 2016). There was one data gap at 11:30 LT on 4 May, 2012 which was linearly interpolated to get continuous time series of wind velocities. High resolution (~4 – 10 m) wind data obtained directly from balloon flights were first sorted in ascending order of height since the balloons occasionally drift downwards by a few meters. The wind profiles were then interpolated vertically to have a constant height resolution of 50 m. This method is useful to smooth the profiles and to maintain a good height resolution to delineate gravity wave parameters. The profiles were then visually inspected for outliers. Only four outliers could be identified out of 20 profiles which were removed and the gaps were filled up by linear interpolation with height.

## 3 Analysis and Discussion

### 3.1 Time series analysis

IGW periods in low latitudes are quite large which makes their observations difficult by using common spectral analysis method. The normal procedure to find the frequency/period of an atmospheric wave is to have a continuous time series data with appropriate data gaps and subject it to Fast Fourier Transform (FFT) technique. The minimum length of data required for FFT analysis is double the period of the wave (Nyquist frequency) to be identified. Keeping this in mind, experiments were conducted as mentioned in section 2 to obtain wind velocities and temperatures continuously for 120 h with a regular interval of 6 h since the IGW period over Hyderabad is ~ 40 h and the data contain three cycles of the wave which satisfies the criterion of FFT technique. This time series data are capable of identifying IGW period after proper filtering and using spectral analysis method. This filtered time series data is considered as reference data for rest of the analyses.

We have used two types of filters. Butterworth filter and Finite Impulse Response (FIR) filter. Butterworth filter belongs to the Infinite Impulse Response (IIR) group of filters. It is a type of signal processing filter designed to have a very flat frequency response in the pass band with a monotonic amplitude response. FIR filters can be reliably designed with linear phase that prevents distortion. These filters can be easily implemented but with the disadvantage that they often require a much higher filter order than IIR filters to achieve a good level of performance. Further details of these filters are available in Butterworth (1930) and Lake (1980). The order of the filter refers to the number of components that affect the steepness or shape of the filter's frequency response. As the order of the filter increases, the cut-off becomes sharper, but the length of the data should be at-least 3 times the filter order. The length of our data is 20 (time-wise) which restricts the maximum order of the filter to be chosen as 6. A Butterworth filter of order 3 is found to be more efficient than a 6$^{th}$ order FIR1 filter for this particular study.

### 3.1.1 Hodograph of wind perturbations using Butterworth filter

The continuous zonal and meridional wind datasets are detrended (linear trend removed) to obtain time series of wind fluctuations. A third order Butterworth filter with a band-pass between 36 and 44 h is applied to the wind perturbations to retrieve the IGW fluctuations with zero phase distortion. The wide band of the time filter is helpful to reduce the Doppler shift of IGW frequency (Niranjan Kumar et al., 2011). Ehard et al., (2015) also recommended the application of Butterworth filter in extracting gravity waves over a wide range of periods from temperature measured by lidar. The filtered horizontal winds at particular heights are depicted in Fig. 1a – 1d which show the presence of IGW with a period of ~ 40 h. FFT analyses carried out with filtered wind fluctuations also reveal the presence of a clear monochromatic wave of the same period (Fig. 1e – 1h) which perhaps, satisfies the requirement of hodograph method.

Hodographs plotted with this time-wise filtered zonal and meridional wind perturbations ($u_{ew}'$, $v_{ns}'$) are quite noisy and it is difficult to identify proper closings. The fluctuation profiles are, therefore, further band-pass filtered using a Butterworth filter with a cut-off at 1.5 – 4 km which produced proper elliptic hodographs. The number of proper hodographs obtained from 20 pairs of vertical profiles of $u_{ew}'$ and $v_{ns}'$ are 124. The polarization relation for internal gravity waves is given by Gubenko et al. (2008, 2011):

$$\frac{v'}{u'} = -i\left(\frac{f}{\omega}\right) \tag{1}$$

where $u'$ and $v'$ are the velocity perturbations for the parallel and perpendicular components of wave-induced horizontal wind relative to the wave propagation direction, correspondingly. This formula implies elliptical wave polarization, with frequency dependent ellipse eccentricity of ($f/\omega$). A few IGW parameters have been extracted using Eq. (1). The horizontal wave number k for internal waves with both low and intermediate intrinsic frequencies ($f^2 < \omega^2 << N^2$) is given by the following dispersion equation (Fritts and Alexander, 2003; Gubenko et al., 2012):

$$|k| = \left(1 - \frac{f^2}{\omega^2}\right)^{1/2} \frac{|m|\omega}{N} \tag{2}$$

where parameters k and m represent the horizontal and vertical wave numbers, $N$ is the Brunt-Väisalä frequency, $f$ and $\omega$ are the inertial (Coriolis parameter) and intrinsic frequencies, correspondingly. Intrinsic periods of IGW obtained using equation (1) from hodographs range between 20 and 28 h which are less than the inertial period for Hyderabad and belong to the intermediate range. The vertical and horizontal wavelengths inferred from the hodographs are between 2.0 to 2.8 km and 569 – 1171 km, respectively.

### 3.1.2. Hodographs using FIR1 filter

We chose a different filter FIR1 of order 6 to test the effect of filtering on hodograph method since the vertical wavelength and intrinsic frequency are reported to be highly vulnerable to the filter used (Zhang et al., 2004). We followed the same procedure to delineate the IGW parameters as described in section 3.1.1. The detrended and time-wise filtered horizontal wind profiles at a few heights and the corresponding FFT peaks are illustrated in Fig. 2a – 2d and 2e – 2h respectively. Both the time variation of wind fluctuations and the FFT peaks do not show distinct IGW periods. The frequency responses of Butterworth filter of 3[rd] order and FIR1 of 6[th] order are shown in Fig. 3. The Butterworth filter shows a sharp cut-off and also requires a much lower filter order than the corresponding FIR1 filter. A few hodographs plotted with horizontal wind perturbations using both the filters are displayed in Fig. 4a – 4d. North is denoted by 0° in the hodographs and its orientation angle increases clockwise. Clockwise rotation of the hodograph indicates upward energy propagation in the northern hemisphere. The IGW parameters derived from these hodographs are listed in Table 1. The ranges of horizontal wavelength, vertical wavelength and intrinsic period are broader using FIR1 filter compared to those obtained using Butterworth filter.

### 3.2. Height series analyses

Hodographs are generally plotted with the fluctuations derived from data of individual sounding by removing polynomials of 1$^{st}$ or 2$^{nd}$ order. We treated the measured vertical profiles of zonal and meridional winds as single individual set (not time series) and approximated the backgrounds by polynomials of different (2 to 9) orders. Fig. 5 depicts different fits and the corresponding wind profiles. The fluctuation profiles obtained by removing polynomials of 4, 5 and 6 orders show good agreements whereas appreciable differences could be noticed for others (figure not shown). These fluctuation profiles are then subjected to different filtering process and hodographs are made. They are subsequently analyzed to derive IGW parameters.

### 3.2.1 Hodographs using Butterworth filter

The perturbation profiles are filtered with a 3$^{rd}$ order Butterworth filter height-wise to retain IGW oscillations with short vertical wavelengths (1.5 – 4 km). IGW parameters obtained from the hodographs plotted with these fluctuations match very well with those described in section 3.1.1

### 3.2.2 Hodographs using FIR1 filter

The individual profiles of winds and temperature are then analyzed in a similar manner as mentioned in section 3.2.1 but by using FIR1 filter with height. The perturbation profiles (after removing backgrounds with different order polynomials) and the filtered fluctuation profiles using both Butterworth and FIR1 filters are shown in Fig. 6a – 6c and 6d – 6f for both the wind components. It can be seen that the Butterworth filter can extract the monochromatic IGW fluctuations very efficiently. The retrieved IGW parameters retain same numerical values (except after decimal points) irrespective of the background removals. Results obtained with FIR1 filter also belong to the same range but with a broader band which is illustrated in Table 2 for different orders.

### 3.3. Direction of wave propagation

The direction of horizontal wave propagation is parallel to the major axis of the $u_{ew}' - v_{ns}'$ hodograph (ellipse) which is uncertain by 180º. This uncertainty can be minimized with the help of additional temperature information. Temperature perturbation profiles are obtained by removing 5$^{th}$ order polynomial fits from the simultaneous temperature profiles and filtering them height-wise with a band-pass Butterworth filter between 1.5 and 4 km. In-phase wind is calculated as Ucosθ where U is the total wind and θ is the corresponding orientation angle of the $u_{ew}' - v_{ns}'$ hodograph (Fig. 4a – 4d). A few hodographs plotted with in-phase winds and temperature fluctuations are illustrated in Fig. 7a – 7d which help in resolving the ambiguity of wave propagation direction (Hu et al., 2002). If the rotation of in-phase wind and temperature perturbation hodograph is clockwise, the direction (angle) of horizontal wave propagation will be the same as the orientation angle determined by $u_{ew}' - v_{ns}'$ hodograph. If the rotation is counter clockwise, it indicates that the propagation direction will be opposite to the orientation angle i.e. orientation angle +180°. As an example, let us consider the hodograph depicted in Fig. 4a. The orientation angle of the major axis of the ellipse is 154.4°. The propagation direction can, therefore, be 154.4° or 154.4°+180°. The corresponding in-phase wind and temperature fluctuation hodograph (Fig. 7a) rotates clockwise confirming the propagation direction to be south-east (154.4°). The unambiguous direction of propagation of IGW is observed to be south-east (58%) in this study. It is necessary to analyze a large number of hodographs to finalize the direction of propagation.

## 4 Summary

Balloon borne experiments have been conducted for five days with an interval of 6 h to characterize IGW using hodograph method. The method is helpful in identifying low-frequency IGW but suffers from several uncertainties. We have utilized the time series of wind fluctuations to extract IGW component by filtering and confirmed it with spectral analysis. Results obtained by using Butterworth and FIR1 filters are compared. A band-pass Butterworth filter with a sharp cut-off is found to isolate the monochromatic IGW component very efficiently. Backgrounds of individual wind profiles have been approximated with polynomials of different orders when the perturbation profiles show reasonable differences. The differences are observed to get reduced when Butterworth filter is used to isolate the IGW components, whereas differences still persist with FIR1 filter. IGW parameters delineated from the corresponding hodographs using the former filter agree extremely well for different order polynomial removals. Results obtained with FIR1 filter also show reasonable agreement but with a broader range. Filtering appears to be of great importance in removing uncertainties of hodograph method. The unambiguous direction of wave propagation can be ascertained using additional and simultaneous temperature information.

## Acknowledgements

Authors are grateful to Indian Space Research Organization (ISRO), Government of India, for providing financial assistance to run the project under its Climate And Weather of Sun–Earth System (CAWSES–II) program. The authors wish to thank India Meteorological Department (IMD), Hyderabad, for their active support to conduct the balloon experiments from their campus. The authors also thank the college management for kind encouragement. Data is available at Vignana Bharathi Institute of Technology, Hyderabad, India. We would like to thank Dr. K. Kishore Kumar, Space Physics Laboratory (SPL) for his valuable discussions. We are grateful to the reviewers for their constructive comments, which helped to improve the paper.

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

**Figure Captions**

**Figure 1.** Time series of filtered (Butterworth filter) fluctuations ($ms^{-1}$) of zonal and meridional winds (a – d) and corresponding FFT spectra (e – f) at a few heights.

**Figure 2.** Same as in Figure 1, but with FIR1 filter.

**Figure 3.** The filter responses of Butterworth (a) and FIR 1(b) filters.

**Figure 4.** Hodographs of horizontal wind fluctuations ($ms^{-1}$) obtained using Butterworth (a, b) and FIR1 (c, d) filters. An open circle and a solid circle in each hodograph indicate the lowest and highest altitudes, respectively. The thin curves represent the elliptical fits.

**Figure 5.** Profiles of zonal and meridional winds ($ms^{-1}$) and their fits with different orders.

**Figure 6. Upper panel:** Vertical profiles of zonal wind fluctuations ($ms^{-1}$) after approximating the backgrounds with different order ($2^{nd}$ – $9^{th}$) polynomials (a) and filtering height-wise with Butterworth filter (b) and FIR1 filter (c). **Lower panel:** Same as upper panel but for meridional wind fluctuations.

**Figure 7.** Hodographs of in-phase wind ($ms^{-1}$) verses temperature fluctuations (K) obtained using Butterworth (a, b) and FIR1 (c, d) filters. An open circle and a solid circle in each hodograph indicate the lowest and highest altitudes, respectively. The thin curves represent the elliptical fits.

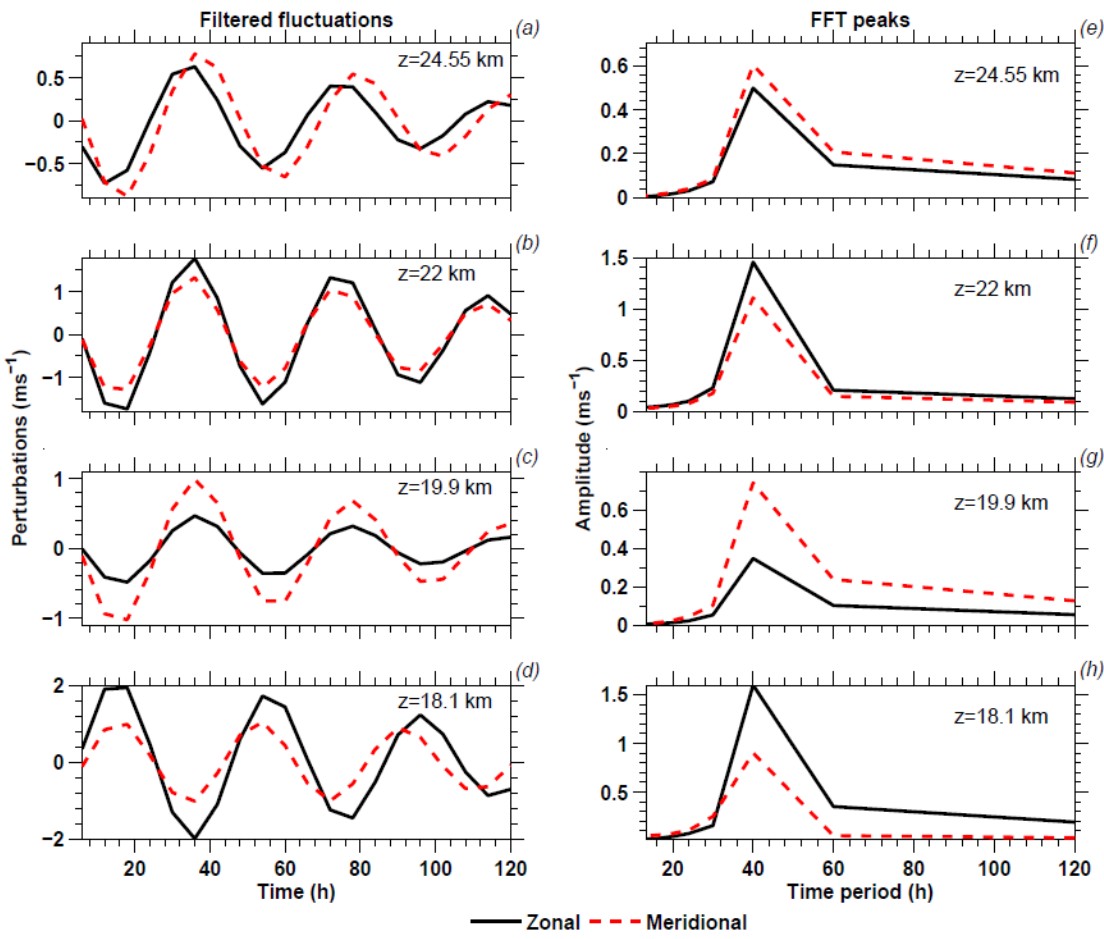

**Figure 1: Time series of filtered (Butterworth filter) fluctuations (ms⁻¹) of zonal and meridional winds (a – d) and corresponding FFT spectra (e – f) at a few heights.**

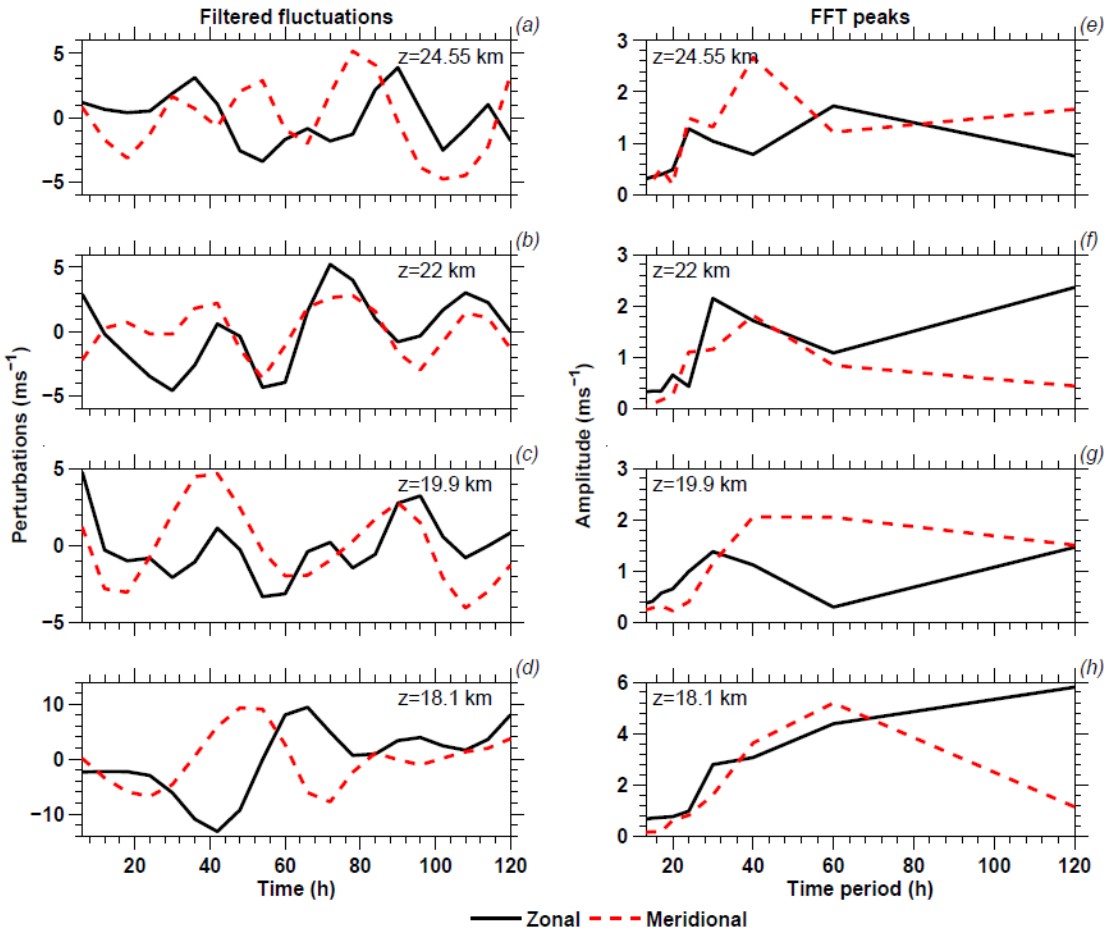

**Figure 2: Same as in Figure 1, but with FIR1 filter.**

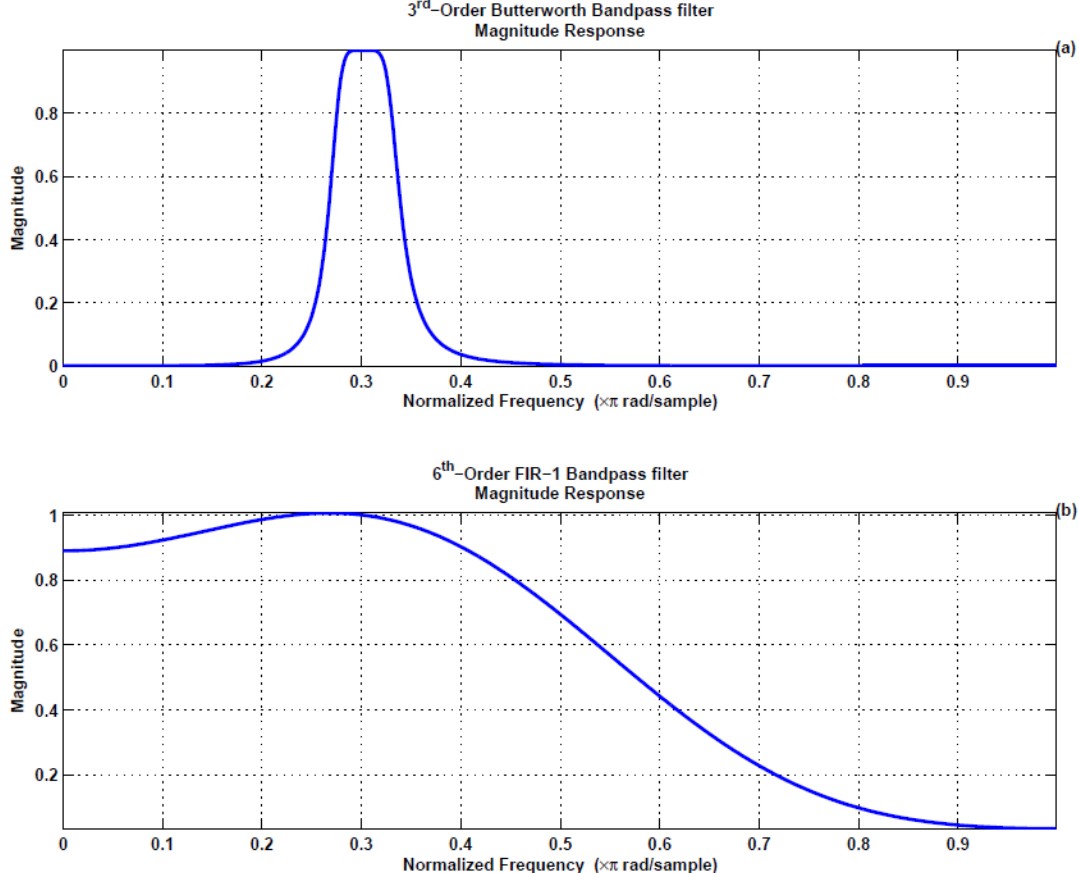

**Figure 3: The filter responses of Butterworth (a) and FIR 1(b) filters.**

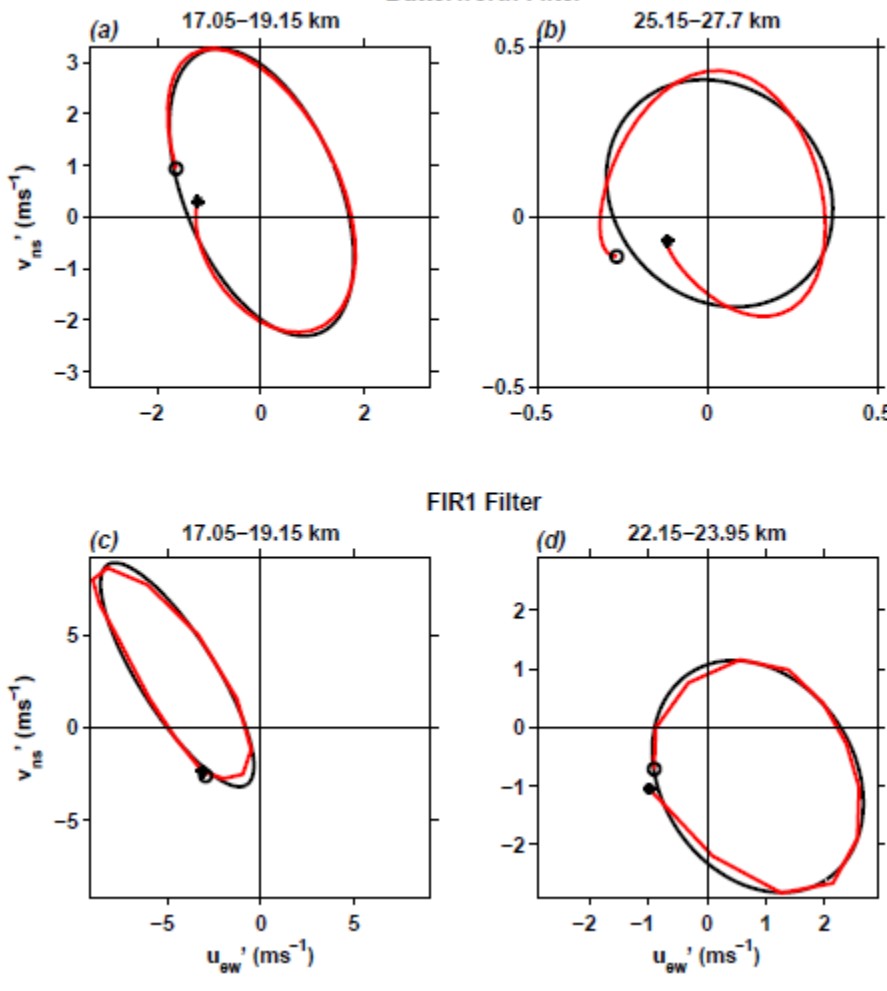

**Figure 4: Hodographs of horizontal wind fluctuations (ms$^{-1}$) obtained using Butterworth (a, b) and FIR1 (c, d) filters. An open circle and a solid circle in each hodograph indicate the lowest and highest altitudes, respectively. The thin curves represent the elliptical fits.**

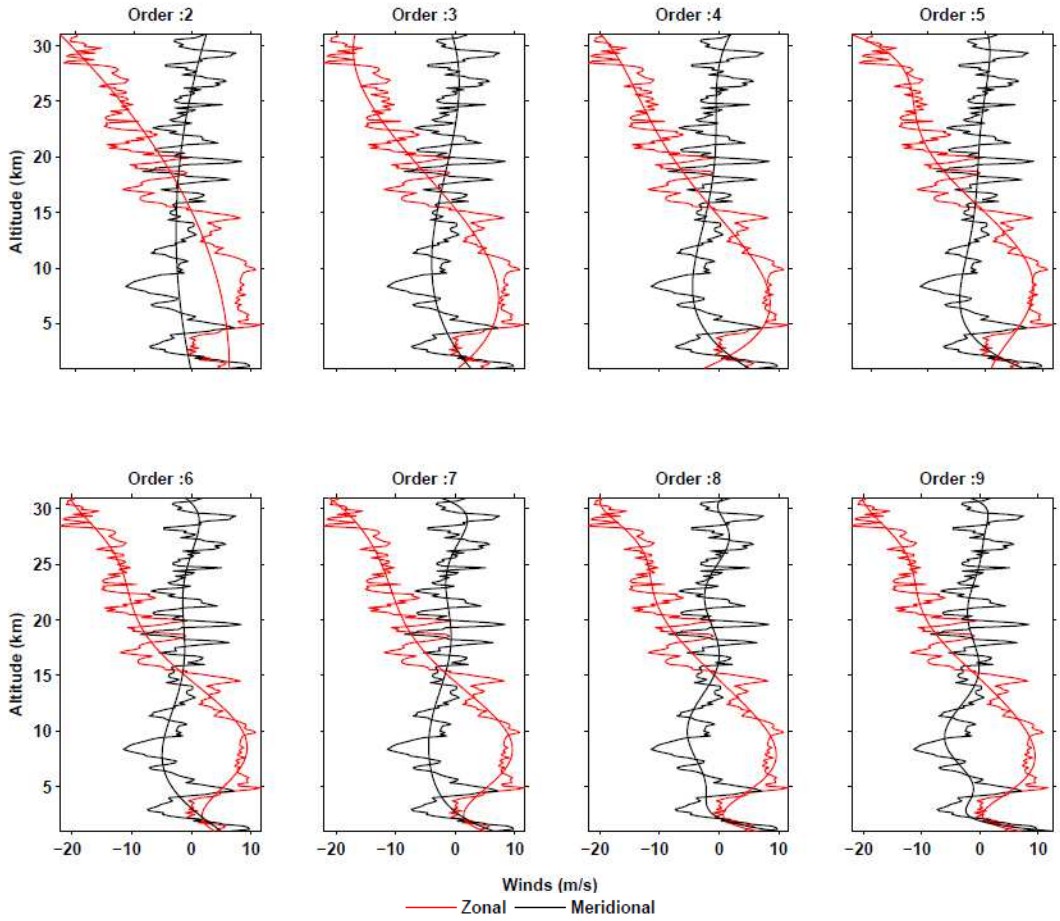

**Figure 5: Profiles of zonal and meridional winds (ms$^{-1}$) and their fits with different orders.**

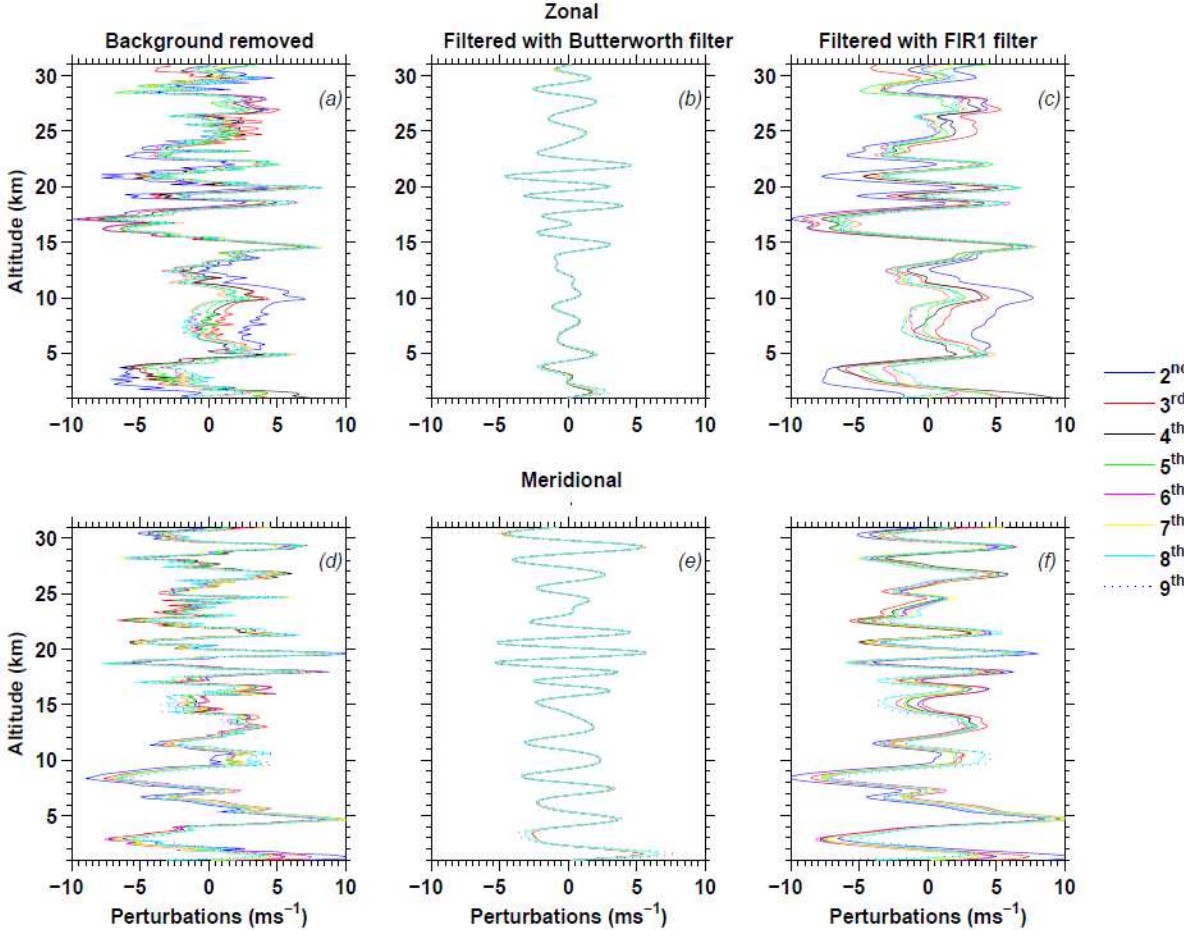

**Figure 6: Upper panel: Vertical profiles of zonal wind fluctuations (ms$^{-1}$) after approximating the backgrounds with different order (2$^{nd}$ – 9$^{th}$) polynomials (a) and filtering height-wise with Butterworth filter (b) and FIR1 filter (c). Lower panel: Same as upper panel but for meridional wind fluctuations.**

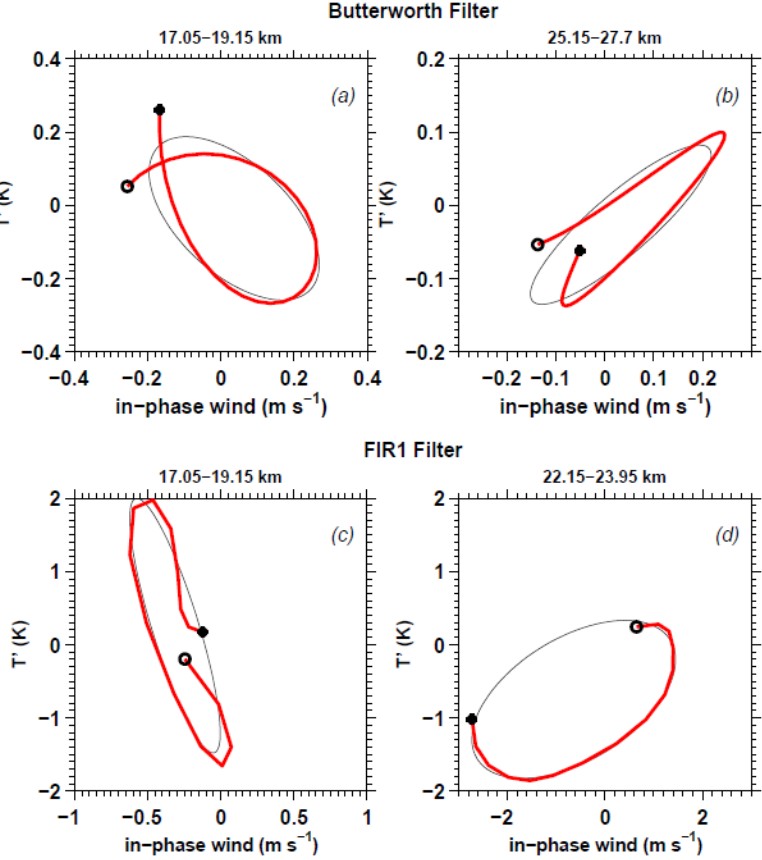

**Figure 7: Hodographs of in-phase wind (ms$^{-1}$) verses temperature fluctuations (K) obtained using Butterworth (a, b) and FIR1 (c, d) filters. An open circle and a solid circle in each hodograph indicate the lowest and highest altitudes, respectively. The thin curves represent the elliptical fits.**

**Table 1:** Comparison of IGW parameters using detrended time series fluctuations and obtained with different filters

| Parameters | Butterworth filter | FIR1 filter |
| --- | --- | --- |
| Horizontal wavelength (km) | 569 – 1171 | 237 – 1209 |
| Vertical wavelength (km) | 2.0 – 2.8 | 1.5 – 3.5 |
| Intrinsic Period (h) | 20 – 28 | 10 – 30 |
| Ratio of minor to major axis | 0.44 – 0.76 | 0.35 – 0.87 |
| Direction of propagation | South-East (58%) | South-East (55%) |

**Table 2:** Comparison of IGW parameters using individual set of wind fluctuation profiles by removing the backgrounds with different order polynomial fits and using both the filters.

| Parameters | | Horizontal wavelength (km) | Vertical wavelength (km) | Intrinsic Period (h) | Ratio of minor to major axis | Direction of propagation |
|---|---|---|---|---|---|---|
| Filter | Order number | | | | | |
| Butterworth | 2 to 9 | 423 – 986 | 2.0 – 2.6 | 16.0 – 25.0 | 0.34 – 0.71 | South – East (52%) |
| FIR1 | 2 | 324 – 882 | 1.7 – 4.0 | 15.0 – 23.0 | 0.34 – 0.71 | South – East (51%) |
| | 3 | 472 – 827 | 1.7 – 4.0 | 17.3 – 23.9 | 0.32 – 0.71 | South – East (58%) |
| | 4 | 404 – 844 | 1.7 – 3.2 | 15.8 – 23.5 | 0.32 – 0.71 | South – East (60%) |
| | 5 | 273 – 1090 | 1.8 – 3.1 | 16.0 – 25.0 | 0.32 – 0.70 | South – East (64%) |
| | 6 | 361 – 905 | 1.7 – 4.0 | 15.8 – 24.7 | 0.30 – 0.69 | South – East (61%) |
| | 7 | 440 – 920 | 1.7 – 4.0 | 16.1 – 25.4 | 0.30 – 0.69 | South – East (56%) |
| | 8 | 360 – 878 | 1.8 – 3.1 | 16.0 – 25.0 | 0.32 – 0.68 | South – East (55%) |
| | 9 | 352 – 739 | 1.7 – 4.0 | 16.2 – 25.0 | 0.31 – 0.68 | South – East (51%) |