# Peer review of "Retrieving characteristics of Inertia Gravity Wave parameters with least uncertainties using hodograph method"

_Atmospheric Chemistry and Physics, 2017_

## Referee Comment (RC1) · Anonymous Referee #1 · 5 Apr 2017

Based on the observational datasets, the authors show the advantage of Butterworth filter in retrieving characteristics of inertial-gravity waves (IGW) with the use of hodograph method. With a very concise storyline, this paper manages to bring relatively new information to the gravity wave community, and it contributes to the application of hodograph method. However, in my view, the structure of the manuscript could be improved, and the authors should try to introduce and justify some of the details in the methodology. For this reason, I would advise MAJOR REVISION. The below paragraphs show my comments in detail, and I believe that this will lead to a very useful and exciting paper once they are addressed.

Major comments:

1. The structure of the manuscript

To the best of my understanding, there are four major groups of experiments in this manuscript. They can be listed as below.

EXP A: 1) Detrend u and v from the time series; 2) Use a third order Butterworth filter with a bandpass between 36 and 44 h, which is a time-wise filter; 3) Use another bandpass filter between 1.5 and 4 km, which is a height-wise filter.

EXP B: The same as EXP A, except that Butterworth filter is replaced by a sixth order FIR1.

EXP C: 1) Obtain the fluctuation profiles by removing polynomial of different orders for each individual profiles. 2) Use a third order Butterworth filter between 1.5 and 4 km, which is a height-wise filter.

EXP D: The same as EXP C, except that Butterworth filter is replaced by FIR1.

Here, by assuming that the IGW characteristics are relatively stationary within 120 h, EXPs A&B could be considered as the reference for EXPs C&D. Also, in reality, EXPs A&B may not be possible due to the requirement of the continuous high-resolution observations in time. In contrast, EXPs C&D are easier to achieve since they only require individual profiles.

The above classification and clarification are summarized by me, and I hope that they are correct. In the current manuscript, it is very hard for the readers to follow the manuscript due to its structure and the lack of the necessary clarification. I would suggest that the methodology part and the list of experiments should be introduced in details in a separate section before the results are shown.

2. The clarification of the details in the methodology

Some of the details in the methodology should be clarified and given. Note that the other reviewer also gave similar comments on an earlier version, but I think that there

is still room for improvement. Please check my below comments.

2.1) On the method of the filter: In addition to Figure 3, the authors should try to present a brief introduction on Butterworth filter and FIR1 filter. Please give the reference on the mathematical calculation of those two filters. Also, what is the meaning of the "order" for each filter? Why is the third order selected for the Butterworth filter? Why is the sixth order selected for the FIR1 filter? Are the results sensitive to the selection of the order?

2.2) Line 49: It seems to me that the measurement errors for wind and temperature could be very close to the wave-induced perturbation of wind and temperature. Please clarify it.

2.3) Lines 52-54: How many outliers or how many data gaps are there? The authors could try to give the ratio of the reliable data versus the interpolated data, if necessary.

2.4) Line 58: In this work, the entire temporal duration is 120 h, and the temporal resolution is 6 h. Therefore, one should be careful about the period under 24 h due to the coarse temporal resolution, and one should also be careful about the period over 60 h due to the assumption of periodic boundary condition. Those similar clarifications should be given. Also, in order to capture a wide range of wave spectrum, it would be nice to have a much higher resolution in time. For example, in Wei et al. (2016, JAS), 1 minute is used as the temporal resolution for the analysis of wave period. This is also worth mentioning.

2.5) Line 84-86: The temperature perturbation profiles are obtained slightly differently from the wind perturbation profiles. Why? Please clarify it.

2.6) In the current study, the authors apply a height-wise bandpass filter (between 1.5 and 4 km) in many calculations. In contrast, Zhang et al. (2004, GRL) actually don't have a height-wise filter. This may be due to the different vertical resolution between the observational studies in the current work and the numerical studies in Zhang et

al. (2004, GRL). The authors should try to clarify those issues related to the above comparison. Is this height-wise filter necessary? What determines the window of the bandpass filter?

2.7) Line 119: I am wondering how to determine the statistical significance with a large number of hodographs? What statistical method is used? What is the minimum sample number required for the significance test? Also, in reality, it may not be possible to have a large number of hodographs.

2.8) Table 1&2: The direction of the propagation is a fixed number. It is strange to me, since the other parameters have a certain range. Please clarify it.

Minor comments:

1. Title: Instead of "IGW", it is better to use "Inertial-Gravity Wave".

2. Line 8: When "IGW" is used for the first time in the abstract, please use its full name.

3. Line 10: When "FIR1" is used for the first time in the abstract (or in the main text), please use its full name.

4. Figure 1: In the subplots, it is better to use "z=24.55 km", instead of "24.55 km". Similarly, please apply it to the other places as well.

5. Figure 6: Please double check the figure caption of Figure 6. (b) should be FIR1 filter, and (c) should be Butterworth filter. The related information is not consistent between figure subtitles and figure caption.

Reference

Wei, J., F. Zhang, and J. H. Richter, 2016: An Analysis of Gravity Wave Spectral Characteristics in Moist Baroclinic Jet-Front Systems. Journal of the Atmospheric Sciences, 73, 3133-3155.

Zhang, F., S. Wang, and R. Plougonven, 2004: Uncertainties in using the hodograph

method to retrieve gravity wave characteristics from individual soundings. Geophysical Research Letters, 31, L11110, doi:10.1029 /2004GL019841.

---

## Referee Comment (RC2) · V. Gubenko (Referee) · 10 Apr 2017

**Referee #2, Vladimir Gubenko**

This paper presents an attempt to overcome the inconsistency of hodograph method when retrieving the internal wave parameters from radisonde measurements. It seems to me that the description of scientific methods and theoretical expressions used for calculations of wave characteristics and their uncertainties **needs to be strongly improved**. For this reason, I would advice **MAJOR REVISION** as the Anonymous referee #1, also. The paper may become suitable for publication in ACP following implementation of the following points.

Major Comments:

1. Page 3, line 75. The values $v'$ and $u'$, in your Eq. 1, **are not** the meridional and zonal wind fluctuations, respectively. The values $u'$ and $v'$ are the complex perturbations for parallel and perpendicular components of wave-induced horizontal wind speed to the wave propagation direction [see for details, for example, Gubenko et al. (2008, JGR, p. 2); Gubenko et al. (2011, AMT, p. 2155); Gubenko et al. (2012, Cosm. Res., p. 22)]. Hu et al. (2002, JRL, p. 1) designate $u'$ as the in-phase wind along the wave propagation direction, and $v'$ as quadrature-phase wind perpendicular to the wave propagation direction.

2. Page 3, line 77. Your Eq. 2 is wrong. The valid expression for the calculation of the inertial frequency $f$ is following (Gubenko et al., 2008, JGR, p. 1). $f = 2\Omega \sin \phi$, where $\Omega = 7.292 \cdot 10^{-5}$rad/s is the Earth's rotation rate, and $\phi$ is latitude.

3. Page 3, line 79. Your Eq. 3 is wrong. In the work of Gubenko et al. (2012, Cosm. Res., p. 23), the dispersion equation in the interval of intermediate intrinsic frequencies ($f^2 \ll \omega^2 \ll N^2$) is given: $|k| = \omega |m| / N$. If we use this expression to calculate the value $|k|$, then calculated values of horizontal wave number $|k|$ will be systematically overestimated by factor $(1 - f^2/\omega^2)^{1/2}$. This is connected with fact that the appropriate dispersion equation that is valid for internal waves with both low and intermediate intrinsic frequencies ($f^2 < \omega^2 \ll N^2$) has form (Gubenko et al. 2012, Cosm. Res., p. 23): $|k| = (1 - f^2/\omega^2)^{1/2} \cdot \omega |m| / N$. For this reason, the obtained results about horizontal wavelengths and wave numbers must be recalculated.

4. Page 3, lines 86–87. You state that the final direction of wave propagation was calculated by using hodographs $u' - v'$ and $u' - t'$ (Hu et al., 2002). I don't understand your method, because Hu et al. (2002, JRL, p. 1) use for that the hodographs of the zonal wind versus meridional wind, and the **in-phase wind** versus temperature.

Minor Comments:

1. Page 2, line 71. For zonal and meridional perturbations it is necessary to introduce another symbols, for example, $u_{we}'$ and $u_{sn}'$

---

## Author Comment (AC2) · 13 Jul 2017

**Referee #2, Vladimir Gubenko**

**Aswers to the comments of Dr. Vladimir Gubenko**

**This paper presents an attempt to overcome the inconsistency of hodograph method when retrieving the internal wave parameters from radiosonde measurements. It seems to me that the description of scientific methods and theoretical expressions used for calculations of wave characteristics and their uncertainties needs to be strongly improved. For this reason, I would advice MAJOR REVISION as the Anonymous referee #1, also. The paper may become suitable for publication in ACP following implementation of the following points.**

**Major Comments:**

1. **Page 3, line 75. The values v' and u', in your Eq. 1, are not the meridional and zonal wind fluctuations, respectively. The values u' and v' are the complex perturbations for parallel and perpendicular components of wave-induced horizontal wind speed to the wave propagation direction [see for details, for example, Gubenko et al. (2008, JGR, p. 2); Gubenko et al. (2011, AMT, p. 2155); Gubenko et al. (2012, Cosm. Res., p. 22)]. Hu et al. (2002, GRL, p. 1) designate u' as the in-phase wind along the wave propagation direction, and v' as quadrature-phase wind perpendicular to the wave propagation direction.**

1A. We have followed the age old method of hodographic analysis. 'u' and 'v' are the profiles of zonal and meridional winds. Height variations of u and v are the profiles of zonal (E – W) and meridional (N – S) velocities only. Please refer to Tsuda et al 1994, (JGR, pg. 10508). "Gravity wave components were extracted…contour plots". Hodographs are plotted with these filtered eastward (zonal wind) and northward (meridional wind) components which are u' and v'. In page 10509, " The lengths of major and minor axes of an ellipse u' - v' correspond to the amplitude of wind velocity fluctuations due to gravity wave", and the formulae follows. Please check the above mentioned paper. The gravity wave fluctuations are normally computed in this manner from measure wind profiles. I am giving you two examples Dutta et al., 2008, JGR and Dutta et al., 2009, JGR. We followed the same procedure to extract gravity wave components.

2. **Page 3, line 77. Your Eq. 2 is wrong. The valid expression for the calculation of the inertial frequency f is following (Gubenko et al., 2008, JGR, p. 1). f = 2Ω sin ϕ, where Ω = 7.292 × 10-5 rad/s is the Earth's rotation rate, and ϕ is latitude.**

2A. Earth's rotation rate (Ω) can be calculated as

$$\Omega = \frac{1}{T} cycles / \sec$$

where 'T' is time period of Earth's rotation (=1day).

According to equation (3) of our paper,

$$f = \frac{\sin \varphi}{\frac{1}{2} day}$$

$$f = \frac{2 \sin \varphi}{1 day}$$

$$f = \frac{2 \sin \varphi}{T}$$

$$f = 2\Omega \sin \varphi \qquad (\because \frac{1}{T} = \Omega)$$

which is the same equation that you have mentioned.

**3.** **Page 3, line 79. Your Eq. 3 is wrong. In the work of Gubenko et al. (2012, Cosm. Res., p. 23), the dispersion equation in the interval of intermediate intrinsic frequencies ($f^2 \ll \omega^2 \ll N^2$) is given: $|k| = \omega |m| / N$. If we use this expression to calculate the value $|k|$, then calculated values of horizontal wave number $|k|$ will be systematically overestimated by factor $(1 - f^2/\omega^2)^{1/2}$. This is connected with fact that the appropriate dispersion equation that is valid for internal waves with both low and intermediate intrinsic frequencies ($f^2 < \omega^2 \ll N^2$) has form (Gubenko et al. 2012, Cosm. Res., p. 23): $|k| = (1 - f^2/\omega^2)^{1/2} \times \omega |m| / N$. For this reason, the obtained results about horizontal wavelengths and wave numbers must be recalculated.**

**3A.** We thank Dr. Gubenko for pointing out this small mistake and giving his important reference paper. We have incorporated the correction factor which improved the quality of the paper.

**4.** **Page 3, lines 86–87. You state that the final direction of wave propagation was calculated by using hodographs u' – v' and u' – t' (Hu et al., 2002). I don't understand your method, because Hu et al. (2002, GRL, p. 1) use for that the hodographs of the zonal wind versus meridional wind, and the in-phase wind versus temperature.**

**4A.** Yes we have done with in-phase wind and temperature. But there is no clarification given in the paper. So now we have incorporated this clarification.

**Minor Comments:**

**1.** **Page 2, line 71. For zonal and meridional perturbations it is necessary to introduce another symbols, for example, $u_{we}'$ and $u_{sn}'$**

**1A.** u' and v' are zonal (E – W) and meridional (N – S) wind fluctuations for us. We use in-phase wind and temperature perturbations in the other hodographs (Figure 7). It is not necessary to introduce the new symbols.

**References:**

Allen, S. J., and R. A. Vincent, 1995: Gravity wave activity in the lower atmosphere: Seasonal and latitudinal variations. J. Geophys. Res., 100, 1327–1350.

Chane-Ming, F., Z. Chen, and F. Roux, 2010: Analysis of gravity waves produced by intense tropical cyclones. Annales Geophysicae, 28, 531–547, doi: 10.5194/angeo-28-531-2010.

Dutta, G., M. C. Ajay Kumar, P. Vinay Kumar, M. Venkat Ratnam, M. Chandrashekar, Y. Shibagaki, M. Salauddin, and H. A. Basha (2009), Characteristics of high-frequency gravity waves generated by tropical deep convection: Case studies, J. Geophys. Res., 114, D18109, doi:10.1029/2008JD011332

Dutta, G., T. Tsuda, P. V. Kumar, M. C. A. Kumar, S. P. Alexander, and T. Kozu (2008), Seasonal variation of short-period (<2 h) gravity wave activity over Gadanki, India (13.5_N, 79.2_E), J. Geophys. Res., 113, D14103, doi:10.1029/2007JD009178.

Hu, X., Liu, A. Z., Gardner, C. S., and Swenson, G. R.: Characteristics of quasi-monochromatic gravity waves observed with Na lidar in the mesopause region at Starfire Optical Range, NM, Geophys. Res. Lett., 29 (24), doi:10.1029/2002GL014975, 2002.

Tsuda, T., Murayama, Y., Wiryosumarto, H., Harijono, S. W. B., and Kato, S.: Radiosonde observations of equatorial atmospheric dynamics over Indonesia, 2. Characteristics of gravity waves. J. Geophys. Res., 99, 10507-10516. 1994.

---

## Author Response (AR1)

**Answers to the reviewers**

Authors thank the reviewers for time and effort spent on evaluating the manuscript and providing suggestions which greatly improved the quality of the paper.

**Anonymous Referee #1**

**Major comments:**

Q1. The structure of the manuscript

To the best of my understanding, there are four major groups of experiments in this manuscript. They can be listed as below.

EXP A: 1) Detrend u and v from the time series; 2) Use a third order Butterworth filter with a bandpass between 36 and 44 h, which is a time-wise filter; 3) Use another bandpass filter between 1.5 and 4 km, which is a height-wise filter.

EXP B: The same as EXP A, except that Butterworth filter is replaced by a sixth order FIR1.

EXP C: 1) Obtain the fluctuation profiles by removing polynomial of different orders for each individual profiles. 2) Use a third order Butterworth filter between 1.5 and 4 km, which is a height-wise filter.

EXP D: The same as EXP C, except that Butterworth filter is replaced by FIR1.

Here, by assuming that the IGW characteristics are relatively stationary within 120 h, EXPs A&B could be considered as the reference for EXPs C&D. Also, in reality, EXPs A&B may not be possible due to the requirement of the continuous high-resolution observations in time. In contrast, EXPs C&D are easier to achieve since they only require individual profiles. The above classification and clarification are summarized by me, and I hope that they are correct. In the current manuscript, it is very hard for the readers to follow the manuscript due to its structure and the lack of the necessary clarification. I would suggest that the methodology part and the list of experiments should be introduced in details in a separate section before the results are shown.

A1. We have taken this valuable comment seriously and attempted to change the matter appropriately, though we have not written the methodology separately. We felt that the results in that case will become somewhat confusing. We request the reviewer to go through this revised portion now and to find out whether clarity is enough.

**Q2. The clarification of the details in the methodology**

Some of the details in the methodology should be clarified and given. Note that the other reviewer also gave similar comments on an earlier version, but I think that there is still room for improvement. Please check my below comments.

2.1 On the method of the filter: In addition to Figure 3, the authors should try to present a brief introduction on Butterworth filter and FIR1 filter. Please give the reference on the mathematical calculation of those two filters. Also, what is the meaning of the "order" for each filter? Why is the third order selected for the

**Butterworth filter? Why is the sixth order selected for the FIR1 filter? Are the results sensitive to the selection of the order?**

2.1 A. A brief introduction of the filters with corresponding references has been incorporated in the matter as per the suggestion of the reviewer.

The order of the filter refers to the number of components that affect the steepness or shape of the filter's frequency response. As the order of the filter increases, the cut-off become sharper, but the length of the data should be at-least 3 times the filter order. The length of our data is 20 (time-wise). So the maximum order of the filter which we could choose is 6. The filter order is normally judiciously chosen by the investigator depending on the efficacy of the filter. A Butterworth filter of order 3 is more efficient than a  $6^{th}$  order FIR1 filter.

- 2.2 Line 49: It seems to me that the measurement errors for wind and temperature could be very close to the wave-induced perturbation of wind and temperature. Please clarify it.
- 2.2 A. The fluctuations are almost of the same order of winds (±10 ms-1) and temperature (±15 K) so the error (mentioned in the paper) is much less compared to the fluctuations
- 2.3 Lines 52-54: How many outliers or how many data gaps are there? The authors could try to give the ratio of the reliable data versus the interpolated data, if necessary.
- 2.3 A. Normally we adopt the method of visual inspection to remove outliers. But in this data we could hardly find 4 small outliers at 4 heights out of 20 profiles with 600 points (heights) each. Only one flight (4th May, 2012; 11:30 LT) data was missing and hence we had to interpolate one point at each height with time.
- 2.4 Line 58: In this work, the entire temporal duration is 120 h, and the temporal resolution is 6 h. Therefore, one should be careful about the period under 24 h due to the coarse temporal resolution, and one should also be careful about the period over 60 h due to the assumption of periodic boundary condition. Those similar clarifications should be given. Also, in order to capture a wide range of wave spectrum, it would be nice to have a much higher resolution in time. For example, in Wei et al. (2016, JAS), 1 minute is used as the temporal resolution for the analysis of wave period. This is also worth mentioning.
- 2.4 A. To avoid this problem, the time series data of 120 h with a gap of 6 h has been filtered between 36 h and 44 h. So waves under 24 h and above 60 h periods are eliminated.

The duration of each radiosonde flight is ~  $1\frac{1}{2}$  h to  $2\frac{1}{2}$  h. It is not possible to fly radiosondes with very high time resolution. Wei et al (2016) is a simulation paper on gravity waves generated by baroclinic instability and it could be possible to take very high time resolution. The work is in the mesosphere which might not be relevant for this work.

**2.5 Line 84-86: The temperature perturbation profiles are obtained slightly differently from the wind perturbation profiles. Why? Please clarify it.**

2.5 A. The velocity and temperature perturbations are normally obtained differently in different papers. We have calculated velocity perturbations by removing different orders of polynomials and we find that removal of 4, 5 and 6 orders yield almost the same results.

Temperature fluctuations have been obtained by removing  $4^{th}$  order polynomial in Hu et al (2002), Allen and Vincent (1995) removed  $2^{nd}$  order polynomial. Chane-Ming et al (2010) removed  $2^{nd}$  and  $3^{rd}$  order polynomial from winds and temperature. No reasons are attributed in any of these papers.

- 2.6 In the current study, the authors apply a height-wise bandpass filter (between 1.5 and 4 km) in many calculations. In contrast, Zhang et al. (2004, GRL) actually don't have a height-wise filter. This may be due to the different vertical resolution between the observational studies in the current work and the numerical studies in Zhang et al. (2004, GRL). The authors should try to clarify those issues related to the above comparison. Is this height-wise filter necessary? What determines the window of the bandpass filter?
- 2.6 A. The height filter is necessary when we analyze individual altitude profiles of winds or temperatures. The vertical wavelength of IGW is short and generally between 2 3 km. The window of the filter is supposed to be selected judiciously by investigator. We have selected it between 1.5 4 km which is commonly taken for IGW studies.

Hodographs plotted with only time wise filtered fluctuations did not yield good hodographs showing some superposition of other waves and hence height wise filtering was needed.

- 2.7 Line 119: I am wondering how to determine the statistical significance with a large number of hodographs? What statistical method is used? What is the minimum sample number required for the significance test? Also, in reality, it may not be possible to have a large number of hodographs.
- 2.7 A. We have not used any statistical significance tests. We have only calculated the percentage of wave propagation in each direction and the maximum number is shown as the final direction of wave propagation. The percentage is mentioned.
- 2.8 Table 1&2: The direction of the propagation is a fixed number. It is strange to me, since the other parameters have a certain range. Please clarify it.
- 2.8 A. Parameters like intrinsic period, horizontal and vertical wavelengths etc are obtained from each of, say, 100 plus hodographs. The values obtained from each hodograph will differ but obviously will be within some range. The maximum and minimum of the ranges have been mentioned.

But the direction of propagation can be NE, SE, SW and NW. The maximum number showing a particular direction is mentioned and the percentage is written. The same is normally followed by other researchers as well.

**Minor comments:**

**1. Title: Instead of "IGW", it is better to use "Inertial-Gravity Wave".**

- A. The word "IGW" in the title has been changed to "Inertia Gravity Wave"
- 2. Line 8: When "IGW" is used for the first time in the abstract, please use its full name.
- A. The full name of IGW has been introduced in the abstract as per the suggestion of the reviewer.
- 3. Line 10: When "FIR1" is used for the first time in the abstract (or in the main text), please use its full name.
- A. The full name of FIR has been introduced in the abstract.
- 4. Figure 1: In the subplots, it is better to use "z=24.55 km", instead of "24.55 km". Similarly, please apply it to the other places as well.
- A. As per the suggestion of the reviewer we have mentioned "z=24.55 km" in the subplots and also applied to other places.
- 5. Figure 6: Please double check the figure caption of Figure 6. (b) should be FIR1 filter, and (c) should be Butterworth filter. The related information is not consistent between figure subtitles and figure caption.
- A. Thanks, figure has been modified accordingly.

**Referee #2, Vladimir Gubenko**

**Aswers to the comments of Dr. Vladimir Gubenko**

This paper presents an attempt to overcome the inconsistency of hodograph method when retrieving the internal wave parameters from radiosonde measurements. It seems to me that the description of scientific methods and theoretical expressions used for calculations of wave characteristics and their uncertainties needs to be strongly improved. For this reason, I would advice MAJOR REVISION as the Anonymous referee #1, also. The paper may become suitable for publication in ACP following implementation of the following points.

**Major Comments:**

1. Page 3, line 75. The values v' and u', in your Eq. 1, are not the meridional and zonal wind fluctuations, respectively. The values u' and v' are the complex perturbations for parallel and perpendicular components of wave-induced horizontal wind speed to the wave propagation direction [see for details, for example, Gubenko et al. (2008, JGR, p. 2); Gubenko et al. (2011, AMT, p. 2155); Gubenko et al. (2012, Cosm. Res., p. 22)]. Hu et al. (2002, GRL, p. 1) designate u' as the in-phase wind along the wave propagation direction, and v' as quadrature-phase wind perpendicular to the wave propagation direction.

- 1A. We have followed the age old method of hodographic analysis. 'u' and 'v' are the profiles of zonal and meridional winds. Height variations of u and v are the profiles of zonal (E W) and meridional (N S) velocities only. Please refer to Tsuda et al 1994, (JGR, pg. 10508). "Gravity wave components were extracted...contour plots". Hodographs are plotted with these filtered eastward (zonal wind) and northward (meridional wind) components which are u' and v'. In page 10509, " The lengths of major and minor axes of an ellipse u' v' correspond to the amplitude of wind velocity fluctuations due to gravity wave fluctuations are normally computed in this manner from measure wind profiles. I am giving you two examples Dutta et al., 2008, JGR and Dutta et al., 2009, JGR. We followed the same procedure to extract gravity wave components.
- 2. Page 3, line 77. Your Eq. 2 is wrong. The valid expression for the calculation of the inertial frequency f is following (Gubenko et al., 2008, JGR, p. 1).  $f = 2\Omega \sin \phi$ , where  $\Omega = 7.292 \times 10-5$  rad/s is the Earth's rotation rate, and  $\phi$  is latitude.
- **2A.** Earth's rotation rate  $(\Omega)$  can be calculated as

$$\Omega = \frac{1}{T} cycles / \sec$$

where 'T' is time period of Earth's rotation (=1day).

According to equation (3) of our paper,

$$f = \frac{\sin \varphi}{\frac{1}{2} day}$$
$$f = \frac{2\sin \varphi}{1day}$$
$$f = \frac{2\sin \varphi}{T}$$
$$f = 2\Omega \sin \varphi \qquad (\because \frac{1}{T} = \Omega)$$

which is the same equation that you have mentioned.

3. Page 3, line 79. Your Eq. 3 is wrong. In the work of Gubenko et al. (2012, Cosm. Res., p. 23), the dispersion equation in the interval of intermediate intrinsic frequencies ( $f^2 << \omega^2 << N^2$ ) is given:  $|\mathbf{k}| = \omega |\mathbf{m}| / \mathbf{N}$ . If we use this expression to calculate the value  $|\mathbf{k}|$ , then calculated values of horizontal wave number  $|\mathbf{k}|$  will be systematically overestimated by factor  $(1 - f^2/\omega^2)^{1/2}$ . This is connected with fact that the appropriate dispersion equation that is valid for internal waves with both low and intermediate intrinsic frequencies ( $f^2 < \omega^2 << N^2$ ) has form (Gubenko et al. 2012, Cosm. Res., p. 23):  $|\mathbf{k}| = (1 - f^2/\omega^2)^{1/2} \times \omega |\mathbf{m}| / \mathbf{N}$ . For this reason, the obtained results about horizontal wavelengths and wave numbers must be recalculated.

- **3A.** We thank Dr. Gubenko for pointing out this small mistake and giving his important reference paper. We have incorporated the correction factor which improved the quality of the paper.
- 4. Page 3, lines 86–87. You state that the final direction of wave propagation was calculated by using hodographs u' v' and u' t' (Hu et al., 2002). I don't understand your method, because Hu et al. (2002, GRL, p. 1) use for that the hodographs of the zonal wind versus meridional wind, and the in-phase wind versus temperature.
- **4A.** Yes we have done with in-phase wind and temperature. But there is no clarification given in the paper. So now we have incorporated this clarification.

**Minor Comments:**

- 1. Page 2, line 71. For zonal and meridional perturbations it is necessary to introduce another symbols, for example,  $u_{we}$ ' and  $u_{sn}$ '
- 1A. u' and v' are zonal (E W) and meridional (N S) wind fluctuations for us. We use inphase wind and temperature perturbations in the other hodographs (Figure 7). It is not necessary to introduce the new symbols.

- 25 convection can coexist together. The popular hodograph method demands the presence of a single coherent wave in the fluctuation profiles and does not yield good result when a mixture of various frequencies are present. The gravity wave

Style Definition: Normal: Font: Times New Roman, 10 pt, English (U.K.), Justified, Space After: 0 pt, Line spacing: 1.5 lines

Style Definition: Header: Font: Times New Roman, 10 pt, English (U.K.), Justified, Space After: 0 pt, Line spacing: 1.5 lines, Tab stops: 8 cm, Centered + 16 cm, Right + Not at 8.25 cm + 16.51 cm

Style Definition: List Paragraph: Font: Times New Roman, 10 pt, English (U.K.), Justified, Space After: 0 pt, Line spacing: 1.5 lines

**Style Definition:** Affiliation: Font: 10 pt, English (U.K.), Justified, Don't add space between paragraphs of the same style

**Style Definition:** Balloon Text: English (U.K.), Justified

Style Definition: Footer: Font: Times New Roman, 10 pt, English (U.K.), Justified, Space After: 0 pt, Line spacing: single, Tab stops: 7.96 cm, Centered + 15.92 cm, Right + Not at 8.25 cm + 16.51 cm

Formatted: Left: 1.65 cm, Right: 1.65 cm, Top: 1 cm, Bottom: 2.36 cm, Width: 21 cm, Height: 24 cm, Header distance from edge: 0 cm, Footer distance from edge: 1.3 cm, From text: 0.4 cm, Numbering: Restart each page

parameters extracted by hodograph method might also be inaccurate when multiple waves are present in the data (Eckermann and Hocking, 1989).

Hodograph method is based on linear theory of gravity waves whereas the dynamics of the flow is more complex and nonlinear which introduces some uncertainties in the interpretation. There are several sources of errors in this method which

- 5 have been described in Zhang et al., (2004). These authors compared the gravity wave characteristics obtained using hodograph method with the values derived from 4D output of their simulation study. A narrow bandwidth filter used by them to extract the fluctuations of a near-monochromatic wave resulted in large uncertainties in the horizontal wavelength which got reduced for waves with shorter vertical wavelengths. Even the spatial variations of the wave characteristics were found to be large. Moreover, since the hodographs are quite variable, a large number of hodographs (profiles) are required to get
- 10 accurate results of gravity wave parameters with some statistical significance (Hall et al., 1995). This defeats the very advantage of the hodograph method which is capable of retrieving GW parameters from a single set of vertical profiles of zonal and meridional winds.

The present paper attempts to overcome the inconsistency of hodograph method in delineating the characteristics of IGW from velocity fluctuations obtained with radiosonde measurements.

**15 2 Experiment and Data**

An intensive campaign with high resolution (i-Met, USA) GPS-radiosonde flights was carried out from the campus of India Meteorological Department (IMD), Hyderabad (17.4 °N, 78.5 °E) with four flights a day at an interval of 6 h for 5 consecutive days (20 flights) between 30 April and 4 May, 2012 to study the characteristics of IGW. The timings of the flights were 05:30, 11:30, 17:30 and 23:30 LT. The accuracy of wind and temperature measurements provided by the 20 manufacturer is  $\pm 1 \text{ ms}^{-1}$  and  $\pm 0.2 \text{ K}$  respectively. There was one data gap at 11:30 LT on 4 May, 2012 which was linearly interpolated to get continuous time series of wind velocities. High resolution ( $\sim 4 - 10$  m) wind data obtained directly from balloon flights were first sorted in ascending order of height since the balloons occasionally drift downwards by a few meters. Wind profiles wereare then visually inspected for outliers and such outliers, if any, wereare removed. The gaps wereare filled up by linear interpolations. The wind profiles wereare then interpolated vertically to have a constant height resolution of 50 m. This method is useful to smooth the profiles and to maintain a good resolution in height.

25

**3 Analysis and Discussion**

Experiments were carried out for five days with a view to getting continuous horizontal wind velocities for 120 h with a regular interval of 6 h keeping in mind that the ICW period over the site is ~40 h.3.1 Time series analysis

IGW periods over low latitudes are quite large which makes their observations difficult by using common spectral analysis

30 method. The normal procedure to find the frequency/period of an atmospheric wave is to have a continuous time series data

with appropriate data gaps and subject it to Fast Fourier Transform (FFT) technique. The minimum length of data required for FFT analysis is double the period of the wave (Nyquist frequency) to be identified. Keeping this in mind, experiments were conducted as mentioned in section 2 to obtain wind velocities and temperatures continuously for 120 h with a regular interval of 6 h since the IGW period over Hyderabad is ~40 h and the data contains three cycles of the wave which satisfies

- the criterion of FFT technique. This time series data is capable of identifying IGW period after proper filtering and using spectral analysis method. The filtered time series data is considered as reference data for rest of the analyses.
   We have used two types of filters. Butterworth filter and Finite Impulse Response (FIR) filter in the present work. Butterworth filter belongs to the Infinite Impulse Response (IIR) group of filters. It is a type of signal processing filter designed to have a very flat frequency response in the pass band with a monotonic amplitude response. FIR filters can be
- 10 reliably designed with linear phase that prevents distortion. These filters can be easily implemented but with the disadvantage that they often require a much higher filter order than IIR filters to achieve a good level of performance. The details of these filters are available in Butterworth (1930) and Lake (1980).

**3.1.1 Hodograph of wind perturbations using Butterwoth filter**

- The continuous zonal and meridional wind datasets are detrended (linear trend removed) to obtain time series of wind
   fluctuations. A third order Butterworth filter with a band-pass between 36 and 44 h is applied to the wind perturbations to
   retrieve the IGW fluctuations with zero phase distortion. The sufficiently wide band of the time filter is helpful to reduce the
   Doppler shift of IGW frequency (Niranjan Kumar et al., 2011). Ehard et al., (2015) also recommended the usage of
   Butterworth filter in extracting gravity waves over a wide range of periods from temperature perturbations measured by
   lidar. The filtered horizontal winds at particular heights are depicted in Fig. 1a 1d which show the presence of IGW with a
   period of ~ 40 h. FFT analyses carried out with filtered wind datafluctuations
- monochromatic wave of the same period (Fig. 1e 1h) which satisfies the requirement of hodograph method.
   Hodographs plotted with this time-wise filtered zonal and meridional wind perturbations (u', v') are found to be quite noisy and it is difficult to identify proper closings. The fluctuation profiles are, therefore, further band-pass filtered using a
- Butterworth filter with a cut-off at 1.5 4 km which produced proper elliptic hodographs. The number of proper hodographs
   obtained from 20 pairs of vertical profiles of u' and v' are 124. A few IGW parameters have been extracted assuming linear dispersion relations (Cho, 1995; Tsuda et al., 1994). The intrinsic wave frequency (ω) is calculated from the ratio of minor to major axes of the ellipse.

$$\frac{v}{w} = -i\left(\frac{f}{\omega}\right)_{\overline{t}}$$

where f is the inertial frequency and  $\frac{v^{t}, u'}{u'}$  are the meridional and zonal amplitudes of wind velocity fluctuations respectively 
[revised manuscript text omitted]

---

## Referee Report (RR1)

**Comment on revised version of "Retrieving characteristics of Inertia Gravity Wave parameters with least uncertainties using hodograph method" by Gopa Dutta et al.**

**Referee #2, Vladimir Gubenko**

This paper presents an attempt to overcome the inconsistency of hodograph method when retrieving the internal wave parameters from radisonde measurements. It seems to me, the description of scientific methods and theoretical expressions used for calculations of wave characteristics and their uncertainties **is not satisfactory and needs to be strongly improved. Authors of this paper should better learn the basic equations and hodograph method for internal gravity waves.** I think that this **paper is not acceptable for publication in the journal with a high standard "Atmospheric Chemistry and Physics"** in its present form. For this reason, I would advice **MAJOR REVISION.**

Major Comments:

1. Page 3, line 27. Authors would like to leave in the text of revised version their polarization Eq. (1) and following designations: $u'$ are zonal wind perturbations and $v'$ are meridional wind perturbations (see author's answers to my comments). Of course, authors may choose any designations for variables, but **in this case your Eq. (1) is not polarization Equation for the values $u'$ and $v'$! In reality, your Eq. (1) is polarization Equation for the parallel and perpendicular perturbation components** of wave-induced horizontal wind speed to the wave propagation direction [see for details, for example, Gubenko et al. (2008, JGR, p. 2); Gubenko et al. (2011, AMT, p. 2155); Gubenko et al. (2012, Cosmic Res., p. 22)].

For the general case of an inertia gravity wave with intrinsic frequency $\omega$, propagating in an atmosphere with Coriolis parameter $f$, the meridional ($v'$) and zonal ($u'$) wind oscillations differ in amplitude and phase, and are related through the following expression (see formula (2) on page 513 of Eckermann and Vincent (1989), or Gossard and Hooke (1975)):

$$v' = \frac{(l/k)\left[1 - i(f/\omega)(k/l)\right]u'}{\left[1 + i(f/\omega)(l/k)\right]} = \frac{(l/k) - i(f/\omega)}{1 + i(f/\omega)(l/k)}u' = \frac{(\omega l - ifk)u'}{\omega k + ifl} \,, \tag{1}$$

where $k$ and $l$ are the zonal and meridional components of the horizontal wavenumber vector, respectively. This formula implies elliptical wave polarization, with frequency dependent ellipse eccentricity of ($f/\omega$). The phase motion of such an inertial gravity wave will have a horizontal component, lying along the major axis of this motion ellipse. **Only in the special case, for a zonally propagating wave ($l = 0$), our polarization Eq. (1) will coincide with polarization Eq. (1) of your manuscript.**

Unfortunately, the citation of Tsuda et al. (1994) is not the real argument, because this work contains the same mistakes as you do.

2. Page 3, line 32. **You have not considered my early comments.** In the work of Gubenko et al. (2012, Cosmic Res., p. 23), the dispersion equation in the interval of intermediate intrinsic frequencies ($f^2 \ll \omega^2 \ll N^2$) is given: $|k| = \omega |m| / N$. This equation assumes **only** linear wave polarization but nor elliptical wave polarization, because $f^2 \ll \omega^2$. **If you use your Eq. (3)** to calculate the value $|k|$, then calculated values of horizontal wave number $|k|$ **will be systematically**

overestimated by factor $(1 - f^2/\omega^2)^{-1/2}$. This is connected with fact that **the appropriate dispersion equation for your case study** should be valid for internal waves both with low and with intermediate intrinsic frequencies ($f^2 < \omega^2 << N^2$). **The appropriate dispersion equation** has form (Gubenko et al. 2012, Cosmic Res., p. 23): $|k| = (1 - f^2/\omega^2)^{1/2} \cdot \omega |m| / N$.

For example, for ratio $f/\omega = 0.6$ (see your Table 2), one can find that $(1 - f^2/\omega^2)^{-1/2} = 1 / 0.8 = 125\%$. Here, we see the significant overestimation by 25% for the calculated value of $|k|$. This is not "small mistake" (see author's answers to my comments), **and the obtained results about horizontal wavelengths and wave numbers must be recalculated.**

3. Page 4, line 5. Your phrase **"Intrinsic periods of IGW obtained using equation (4) …"** is completely incomprehensible.

4. The description **of Section 3.3 "Direction of wave propagation" is not satisfactory,** and it should be remade.

**References**

Eckermann, S.D. and Vincent, R.A., 1989. Falling Sphere Observation of Anisotropic Gravity Wave Motions in the Upper Stratosphere over Australia. Pure Appl. Geophys. 130 (2/3), 509–532.

Gossard, E.E. and Hooke, W.H., 1975. Waves in the Atmosphere, Elsevier, Amsterdam.

Gubenko, V.N., Pavelyev, A.G., Andreev, V.E., 2008. Determination of the intrinsic frequency and other wave parameters from a single vertical temperature or density profile measurement. J. Geophys. Res. 113, D08109, doi:10.1029/2007JD008920.

Gubenko, V.N., Pavelyev, A.G., Salimzyanov, R.R., Pavelyev, A.A., 2011. Reconstruction of internal gravity wave parameters from radio occultation retrievals of vertical temperature profiles in the Earth's atmosphere. Atmos. Meas. Tech. 4, 2153–2162, doi: 10.5194/amt-4-2153-2011.

Gubenko, V.N., Pavelyev, A.G., Salimzyanov, R.R., Andreev, V.E., 2012. A method for determination of internal gravity wave parameters from a vertical temperature or density profile measurement in the Earth's atmosphere. Cosmic Res. 50, 21–31, doi: 10.1134/S0010952512010029.

Tsuda, T., Murayama, Y., Wiryosumarto, H., Harijono, S.W.B., Kato, S., 1994. Radiosonde observations of equatorial atmospheric dynamics over Indonesia, 2. Characteristics of gravity waves. J. Geophys. Res., 99, 10507-10516.

---

## Referee Report (RR2)

**Comment on revised version of "Retrieving characteristics of Inertia Gravity Wave parameters with least uncertainties using hodograph method" by Gopa Dutta et al.**

**Referee #2, Vladimir Gubenko**

This paper presents an attempt to overcome the inconsistency of hodograph method when retrieving the internal wave parameters from radisonde measurements. The paper may become suitable for publication in ACP following implementation of the following points.

Major Comments:

1. Text from line 13, page 3 ("For the general case…") to line 27, page 3 ("…obtained using equation (2)") should be deleted. Instead of this text it is need to write the following:

"The polarization relation for internal gravity waves is given by Gubenko et al. (2008, 2011):

$$v'/u' = -i\,(f/\omega), \tag{1}$$

where $u'$ and $v'$ are the velocity perturbations for the parallel and perpendicular components of wave-induced horizontal wind relative to the wave propagation direction, correspondingly. This formula implies elliptical wave polarization, with frequency dependent ellipse eccentricity of $(f/\omega)$. A few IGW parameters have been extracted using Eq. (1). The horizontal wave number $k$ for internal waves with both low and intermediate intrinsic frequencies ($f^2 < \omega^2 << N^2$) is given by the following dispersion equation (Fritts and Alexander, 2003; Gubenko et al., 2012):

$$|k| = (1 - f^2/\omega^2)^{1/2} \cdot \omega\,|m|\,/\,N, \tag{2}$$

where parameters $k$ and $m$ represent the horizontal and vertical wave numbers, $N$ is the Brunt-Vaisala frequency, $f$ and $\omega$ are the inertial (Coriolis parameter) and intrinsic frequencies, correspondingly. Intrinsic periods of IGW obtained using equation (1)"

P.S.: Because you agree to introduce $u_{we}'$ and $u_{sn}'$, these corrections are necessary to avoid contradictions with the designations.

**References**

Fritts, D.C., and Alexander M.J.: Gravity wave dynamics and effects in the middle atmosphere, Rev. Geophys., 41(1), 1003, doi:10.1029/ 2001RG000106, 2003.

Gubenko, V.N., Pavelyev, A.G., Andreev, V.E., 2008. Determination of the intrinsic frequency and other wave parameters from a single vertical temperature or density profile measurement. J. Geophys. Res. 113, D08109, doi:10.1029/2007JD008920.

Gubenko, V.N., Pavelyev, A.G., Salimzyanov, R.R., Pavelyev, A.A., 2011. Reconstruction of internal gravity wave parameters from radio occultation retrievals of vertical temperature profiles in the Earth's atmosphere. Atmos. Meas. Tech. 4, 2153–2162, doi: 10.5194/amt-4-2153-2011.

Gubenko, V.N., Pavelyev, A.G., Salimzyanov, R.R., Andreev, V.E., 2012. A method for determination of internal gravity wave parameters from a vertical temperature or density profile measurement in the Earth's atmosphere. Cosmic Res. 50, 21–31, doi: 10.1134/S0010952512010029.

---

## Editor Decision (ED1)

Thank you for revising the manuscript again. This is a much improved version. However, there are still certain things to be clarified for making a publication on ACP. Please consider the referee comments and the following to revise your manuscript, and submit it.

**Page 1**

Line 06:  We have analyzed wind velocities measured …

Lin 06: GPS acronym should be expanded.

Line 07: GPS radiosonde, which is continuously flown for 120 h with an interval of 6 h

Line 09: to get the fluctuations from measurements?  What are these fluctuations?

Line 13: acceptable results? How do we know that these are acceptable or not?

Line 13: The FIR1 filter also…

Line 17: delete large

Line 26: "are present", where?

Line 30: uncertainties in their calculations or estimates

Line 30: for errors or error sources

**Page 2**

Line 01:  This limits (not defeats)

Line 02: of hodograph

Line 02: in fact "the method can be used", not capable of doing it

Line 04: In this study, we attempt to reduce the uncertainties associated with…., write something like this.

Line 04: you cannot overcome, but minimize

Line 05: The instrument measures wind velocity. You compute the perturbations or fluctuations afterwards

Line 10—11: remove "provided by the manufacture" and then give a reference.

Line 15: interpolation / smoothing will not give you high resolution. You are just interpolating the values in between and that's all.

Line 21: in low latitudes

Line 23: continuous data with appropriate data gaps? Why?

Line 26: contain

Line 27: data are

Line 29: delete "in the present work"

Line 34: Further details of these filters

Line 37—38: "A Butterworth……" for this particular study, not in general.

**Page 3**

Line 04: "the wide band of"

Line 05: recommended the application of

Line 06: temperature measured by

Line 09: Perhaps, satisfies the criteria for applying the hodograph method

Line 10: "are quite noisy"

Line 24: space after the bracket

Line 27: between 20 and 28 h

Line 28: for Hyderabad and belong to

Line 29: ,respectively

Line 31: delete "Next"

Line 33: delete "but by using…filter"

Line 37: "producing good result"? How do we know that these results are good? Please justify with a relevant sentence/statement.

**Page4:**

Line 02: are broader

Line 06: depicts different fits

Line 07—08: show good agreement

Line 08: what are appreciable differences?

Line 09: "and hodographs are made."

Line 09: subsequently not consequently

Line 14: very, not extremely

Line 16: delete: "instead of …filter."

Line 19: for both wind components. If you use "respectively", then you need to write the "respective"components too.

Line 19: "It is shown that", remove clear. Let the readers decide whether this is clear to them or not.

Line 25: you can only reduce or minimize the uncertainty not remove it completely.

**Page 5**

Line 01: five-day balloon measurements are OK for characterizing IGWs?

Please make a general statement on the measurements here. Characterization comes afterwards. People could also use the data for other purposes (not only for identifying IGWs studies).

Line 31: comma after Australia

Line 37: space before source

Page6

Line 03, 05, 07, etc..hyphen not minus sign

Line 29: comma after systems

Line 36: comma before but

These are some examples for language /syntax corrections. Please read and do other corrections. Thank you.

---

## Author Response (AR2)

**Answers to Reviewers' Comments**

**Referee #1,**

I have checked the response and the revised manuscript. To me, the reply message is clear and informative. My overall impression is that the revised manuscript is improved. The previous main concerns have been widely addressed. However, in my opinion, there may be just three minor issues that could be addressed.

**Minor comments**

1. The reply message (2.1 A.) shows the justification of the order selected for different filters (e.g., the third order selected for the Butterworth filter, the sixth order selected for the FIR1 filter), but this information is not indicated in the revised manuscript. As far as I am concerned, it is better to include this message in the revised manuscript.

**Reply. As per the suggestion of the reviewer, we have incorporated the justification of the selection of order in the revised manuscript.**

2. Similar to the above comment, it is also better to include the reply message of 2.3 A. in the revised manuscript, which is about the clarification of outliers or data gaps.
**Reply. We have incorporated the clarification of outliers and data gaps in the revised manuscript.**

3. According to the reply message in 2.7 A., the statistical significance test is not used in this paper, and the authors haven't yet introduced the proposals about how

to determine the statistical significance with a large number of hodographs. Therefore, I would suggest that the authors delete or modify the last sentence of the summary section, in order to eliminate the redundant information and potential confusion.

**Reply. The last sentence from abstract, section 3.3 and in the summary section "but a large number of hodographs are needed to confirm it with statistical significance" have been deleted in the revised manuscript.**

**Comment on revised version of "Retrieving characteristics of Inertia Gravity Wave parameters with least uncertainties using hodograph method" by Gopa Dutta et al.**

**Referee #2, Vladimir Gubenko**

This paper presents an attempt to overcome the inconsistency of hodograph method when retrieving the internal wave parameters from radiosonde measurements. It seems to me, the description of scientific methods and theoretical expressions used for calculations of wave characteristics and their uncertainties **is not satisfactory and needs to be strongly improved**. **Authors of this paper should better learn the basic equations and hodograph method for internal gravity waves.** I think that this **paper is not acceptable for publication in the journal with a high standard "Atmospheric Chemistry and Physics"** in its present form. For this reason, I would advice **MAJOR REVISION.**

Major Comments:

1. Page 3, line 27. Authors would like to leave in the text of revised version their polarization Eq. (1) and following designations: *u'* are zonal wind perturbations and *v'* are meridional wind perturbations (see author's answers to my comments). Of course, authors may choose any designations for variables, but **in this case your Eq. (1) is not polarization Equation for the values *u'* and *v'*! In reality, your Eq. (1) is polarization Equation for the parallel and perpendicular perturbation components** of wave-induced horizontal wind speed to the wave propagation direction [see for details, for example, Gubenko et al. (2008, JGR, p. 2); Gubenko et al. (2011, AMT, p. 2155); Gubenko et al. (2012, Cosmic Res., p. 22)].

For the general case of an inertia gravity wave with intrinsic frequency ω, propagating in an atmosphere with Coriolis parameter $f$, the meridional ($v'$) and zonal ($u'$) wind oscillations differ in amplitude and phase, and are related through the following expression (see formula (2) on page 513 of Eckermann and Vincent (1989), or Gossard and Hooke (1975)):

$$v' = \frac{(l/k)[1 - i(f/\omega)(k/l)]u'}{[1 + i(f/\omega)(l/k)]} = \frac{(l/k) - i(f/\omega)}{1 + i(f/\omega)(l/k)}u' = \frac{(\omega l - ifk)}{\omega k + ifl}u' \qquad (1)$$

where $k$ and $l$ are the zonal and meridional components of the horizontal wavenumber vector, respectively. This formula implies elliptical wave polarization, with frequency dependent ellipse eccentricity of ($f/\omega$). The phase motion of such an inertial gravity wave will have a horizontal component, lying along the major axis of this motion ellipse. **Only in the special case, for a zonally propagating wave ($l = 0$), our polarization Eq. (1) will coincide with polarization Eq. (1) of your manuscript.**

Unfortunately, the citation of Tsuda et al. (1994) is not the real argument, because this work contains the same mistakes as you do.

**Reply: Following the referee's suggestion, we have gone through the seminal studies cited by him on IGW. Now, we understood our mistake in depicting the polarization equation (equation 1 in the manuscript), which correspond to special case i.e., when l=0. We have made changes accordingly in the revised manuscript.**

2. Page 3, line 32. **You have not considered my early comments.** In the work of Gubenko et al. (2012, Cosmic Res., p. 23), the dispersion equation in the interval of intermediate intrinsic frequencies ($f^2 << \omega^2 << N^2$) is given: $|k| = \omega |m| / N$. This

equation assumes **only** linear wave polarization but nor elliptical wave polarization, because $f^2 \ll \omega^2$. **If you use your Eq. (3)** to calculate the value $|k|$, then calculated values of horizontal wave number $|k|$ **will be systematically overestimated by factor $(1 - f^2/\omega^2)^{-1/2}$**. This is connected with fact that **the appropriate dispersion equation for your case study** should be valid for internal waves both with low and with intermediate intrinsic frequencies ($f^2 < \omega^2 \ll N^2$). **The appropriate dispersion equation** has form (Gubenko et al. 2012, Cosmic Res., p. 23): $|k| = (1 - f^2/\omega^2)^{1/2} \cdot \omega |m| / N$.

For example, for ratio $f/\omega = 0.6$ (see your Table 2), one can find that $(1 - f^2/\omega^2)^{-1/2}$ = 1 / 0.8 = 125%. Here, we see the significant overestimation by 25% for the calculated value of $|k|$. This is not "small mistake" (see author's answers to my comments), **and the obtained results about horizontal wavelengths and wave numbers must be recalculated.**

**Reply: We had actually used the proper equations suggested by the Dr. Gubenko. But we have also retained equation 3. Now we have removed equation 3. There is no over estimation of the parameters in the tables since we have calculated them by using equation 3 of the revised manuscript. We request the reviewer to check our earlier manuscript where we have calculated parameters using equation k=mω/N and the values projected in this manuscript and the previous one are already corrected.**

3. Page 4, line 5. Your phrase **"Intrinsic periods of IGW obtained using equation (4) …"** is completely incomprehensible.

**Reply: We thank the reviewer for projecting this mistake. We have corrected it in the revised manuscript.**

4. The description **of Section 3.3 "Direction of wave propagation" is not satisfactory,** and it should be remade.

**Reply: As per the suggestion of the reviewer we have clarified the direction of wave propagation in the revised manuscript.**

**Overall, we thank Dr. Gubenko for giving strong comments which helped to improve the quality of the paper reasonably.**

**References**

[revised manuscript text omitted]

---

## Author Response (AR3)

**ANSWERS TO THE COMMENTS**

**Thank you for revising the manuscript again. This is a much improved version. However, there are still certain things to be clarified for making a publication on ACP. Please consider the referee comments and the following to revise your manuscript, and submit it.**

Authors thank the co-editor and reviewer for time and effort spent on evaluating the manuscript and providing suggestions which greatly improved the quality of the paper.

**ANSWERS TO CO-EDITOR'S COMMENTS**

**Page 1**

**Line 06: We have analyzed wind velocities measured …**

    A. "time series of" has been removed from the text now. (See line 06)

**Lin 06: GPS acronym should be expanded.**

    A. Acronym for GPS has been given now. (See line 06)

**Line 07: GPS radiosonde, which is continuously flown for 120 h with an interval of 6 h**

    A. The sentence has been re-written as "We have analyzed wind velocities measured with high resolution Global positioning System (GPS) radiosondes which have been flown continuously for 120 h with an interval of 6 h from Hyderabad." (See lines 6 – 7)

**Line 09: to get the fluctuations from measurements? What are these fluctuations?**

    A. Backgrounds are supposed to be removed to obtain fluctuations from time series or height series. For example, if it is a time series of winds, then mean of the time series is removed from each individual observation to get the fluctuations.

    When we consider the height series of the same parameter, polynomials are removed to obtain the fluctuations.

**Line 13: acceptable results? How do we know that these are acceptable or not?**

    A. "Acceptable" – we meant the results are comparable with the other reports. Now we have deleted the word "acceptable". (See line 13)

**Line 13: The FIR1 filter also…**

    A. "The" has been added before "FIR1 filter also…" now. (See line 13)

**Line 17: delete large**

    A. The word "large" has been deleted now. (See line 17)

**Line 26: "are present", where?**

A. When number of waves are present in the data which requires proper filtering to retain the monochromatic wave needed for the particular study.

**Line 30: uncertainties in their calculations or estimates**

A. Uncertainties are in the estimates.

**Line 30: for errors or error sources**

A. Error sources (Please refer to paper by Zhang et al., 2004).

**Page 2**

**Line 01: This limits (not defeats)**

A. The word "defeats" has been replaced with "limits". (See line 1)

**Line 02: of hodograph**

A. The word "the" has been removed. (See line 2)

**Line 02: in fact "the method can be used", not capable of doing it**

A. "capable of retrieving" has been changed to "used to retrieve". (See line 2)

**Line 04: In this study, we attempt to reduce the uncertainties associated with…., write something like this.**

**Line 04: you cannot overcome, but minimize**

A. The sentence has been modified as "In this study, we have attempted to reduce uncertainties associated with hodograph method in delineating the characteristics of IGW from wind velocities obtained with radiosonde measurements." (See lines 4 – 5)

**Line 05: The instrument measures wind velocity. You compute the perturbations or fluctuations afterwards**

A. "velocity fluctuations" has been changed as" wind velocities". (See line 5)

**Line 10—11: remove "provided by the manufacture" and then give a reference.**

A. "provided by the manufacture" has been removed and a reference has been given. The reference is also added in the reference list now. (See lines 10 – 11)

**Line 15: interpolation / smoothing will not give you high resolution. You are just interpolating the values in between and that's all.**

A. The resolution of the radiosonde data is very high (~4 – 10 m) which may contain some noise as well. Interpolation with a resolution of 50 m helps us to reduce the noise and smooth the profile. The resolution of 50 m is good enough to extract gravity wave parameters. If you permit, we would like to retain this sentence. The sentence has now

been modified as "This method is useful to smooth the profiles and to maintain a good height resolution to delineate gravity wave parameters." (See lines 14 – 15)

**Line 21: in low latitudes**

A. The word "over" has been replaced with "in". (See line 20)

**Line 23: continuous data with appropriate data gaps? Why?**

A. IGW over Hyderabad is ~ 40 h. We would like to have a data gap of 5 h which is an exact fraction of 40. But the experiment was a part of a campaign at national level and we have to adjust the timings of flights for a gap of 6 h. The data gap (6 h) is enough to observe IGW with good accuracy by FFT.

**Line 26: contain**

A. The word "contains" has been replaced with "contain". (See line 25)

**Line 27: data are**

A. The word "is" is replaced with "are". (See line 26)

**Line 29: delete "in the present work"**

A. "in the present work" has been deleted. (See line 28)

**Line 34: Further details of these filters**

A. The word "The" has been replaced with "Further". (See lines 32)

**Line 37—38: "A Butterworth……" for this particular study, not in general.**

A. "for this particular study" has been added now. (See line 36 – 37)

**Page 3**

**Line 04: "the wide band of"**

A. The word "sufficiently" is deleted. (See line 4)

**Line 05: recommended the application of**

A. The word "usage" has been replaced by "application". (See line 5)

**Line 06: temperature measured by**

A. The word "perturbations" has been deleted. (See line 6)

**Line 09: Perhaps, satisfies the criteria for applying the hodograph method**

A. The word "perhaps," has been added. (See line 9)

**Line 10: "are quite noisy"**

A. "found to be" has been removed. (See line 10)

**Line 24: space after the bracket**

A. Space has been given now. (See line 19)

**Line 27: between 20 and 28 h**

A. "20 – 28 h" has been replaced with 20 and 28 h. (See line 24)

**Line 28: for Hyderabad and belong to**

A. "of Hyderabad and belongs to" has been changed to "for Hyderabad and belong to". (See line 24)

**Line 29: , respectively**

A. A comma has been added before "respectively". (See line 26)

**Line 31: delete "Next"**

A. "Next" has been deleted. (See line 28)

**Line 33: delete "but by using…filter"**

A. "but by using…filter" has been deleted. (See line 30)

**Line 37: "producing good result"? How do we know that these results are good? Please justify with a relevant sentence/statement.**

A. The sentence has been modified as "The Butterworth filter shows a sharp cut-off and also requires a much lower filter order than the corresponding FIR1 filter." (See line 33 – 34)

**Page4:**

**Line 02: are broader**

A. It has been incorporated. (See page 3, line 38)

**Line 06: depicts different fits**

A. "the" has been deleted. (See line 4)

**Line 07—08: show good agreement**

A. "close" has been replaced with "good". (See lines 5 – 6)

**Line 08: what are appreciable differences?**

A. Fluctuation profiles by removing $2^{nd}$, $4^{th}$, $5^{th}$ and $9^{th}$ order polynomials from original profile are shown together (see figure below). It clearly shows that fluctuations obtained by removing $4^{th}$ and $5^{th}$ orders match quite well whereas the other two do not.

[Figure]

Figure: Vertical profiles of wind velocity fluctuations for different order polynomial fit removals.

**Line 09: "and hodographs are made."**

    A. "plotted" has been changed to "made". (See line 7)

**Line 09: subsequently not consequently**

    A. "consequently" has been changed to "subsequently". (See line 7)

**Line 14: very, not extremely**

    A. "extremely" has been changed to "very". (See line 11)

**Line 16: delete: "instead of …filter."**

    A. "instead of …filter" has been deleted. (See line 15)

**Line 19: for both wind components. If you use "respectively", then you need to write the "respective" components too.**

    A. The word "respectively" has been deleted. (See line 17)

**Line 19: "It is shown that", remove clear. Let the readers decide whether this is clear to them or not.**

    A. "It is clearly observed" has been changed to "It can be seen that". I request the editor to agree with this change. (See line 17)

**Line 25: you can only reduce or minimize the uncertainty not remove it completely.**

    A. The word "removed" has been changed to "minimized". (See line 23)

**Page 5**

**Line 01: five-day balloon measurements are OK for characterizing IGWs?**

**Please make a general statement on the measurements here. Characterization comes afterwards. People could also use the data for other purposes (not only for identifying IGWs studies).**

    A. The data gap of 6 h and the total duration (5 days) have been mentioned which is sufficient information for people working with waves to know what kinds of waves can be studied with such data. We have only done the experiment to study IGW and characterize it with Hodograph method. I, therefore, request you to retain the summary as it is.

**Line 31: comma after Australia**

    A. A comma has been place after "Australia". (See line 31)

**Line 37: space before source**

    A. A space has been given before "Source". (See line 37)

**Page6**

**Line 03, 05, 07, etc., hyphen not minus sign**

    A. Minus sign has been replaced with hyphen wherever it is appropriate.

**Line 29: comma after systems**

    A. A comma has been place after "systems". (See line 37)

**Line 36: comma before but**

    A. A comma has been place before "but". (See page 7, line 4)

**These are some examples for language /syntax corrections. Please read and do other corrections. Thank you.**

The paper has been thoroughly read and the corrections have been made as per the editor's suggestion.

**Answer to referee #2, Vladimir Gubenko:**

The equations and sentences have been changed as per the suggestion of the reviewer.

(See page 3, lines 13 – 24)